# Callus Culture of Thai Basil Is an Effective Biological System for the Production of Antioxidants

**DOI:** 10.3390/molecules25204859

**Published:** 2020-10-21

**Authors:** Saher Nazir, Hasnain Jan, Duangjai Tungmunnithum, Samantha Drouet, Muhammad Zia, Christophe Hano, Bilal Haider Abbasi

**Affiliations:** 1Department of Biotechnology, Quaid-i-Azam University, Islamabad 45320, Pakistan; saher_nazir158@yahoo.com (S.N.); rhasnain849@gmail.com (H.J.); ziachaudhary@gmail.com (M.Z.); 2Department of Pharmaceutical Botany, Faculty of Pharmacy, Mahidol University, Bangkok 10400, Thailand; duangjai.tun@mahidol.ac.th; 3Laboratoire de Biologie des Ligneux et des Grandes Cultures, INRAE USC1328, University of Orleans, CEDEX 02, 45067 Orléans, France; samantha.drouet@univ-orleans.fr

**Keywords:** antioxidants, biological system, caffeic acid, callus culture, chicoric acid, phenolics, plant in vitro cultures, rosmarinic acid, thai basil

## Abstract

Thai basil is a renowned medicinal plant and a rich source of bioactive antioxidant compounds with several health benefits, with actions to prevent of cancer, diabetes and cardiovascular disease. Plant cell and tissue culture technologies can be routinely established as an important, sustainable and low-cost biomass source to produce high-value phytochemicals. The current study aimed at developing an effective protocol to produce Thai basil leaf-derived callus cultures with sustainable and high production of biomass and antioxidants as an alternative of leaves production. MS basal medium with various concentrations of plant growth regulators (PGRs) compatible with nutraceutical applications (i.e., gibberellic acid (GA_3_) and 6-benzylaminopurine (BAP) either alone or in combination with naphthalene acetic acid (NAA)) were evaluated. Among all tested PGRs, the combination BAP:NAA (5 mg/L:1 mg/L) yields the maximum biomass accumulation (fresh weight (FW): 190 g/L and dry weight (DW): 13.05 g/L) as well as enhanced phenolic (346.08 mg/L) production. HPLC quantification analysis indicated high productions of chicoric acid (35.77 mg/g DW) and rosmarinic acid (7.35 mg/g DW) under optimized callus culture conditions. Antioxidant potential was assessed using both in vitro cell free and in vivo cellular antioxidant assays. Maximum in vitro antioxidant activity DPPH (93.2% of radical scavenging activity) and ABTS (1322 µM Trolox equivalent antioxidant capacity) was also observed for the extracts from callus cultures grown in optimal conditions. In vivo cellular antioxidant activity assay confirmed the effective protection against oxidative stress of the corresponding extract by the maximum inhibition of ROS and RNS production. Compared to commercial leaves, callus extracts showed higher production of chicoric acid and rosmarinic acid associated with higher antioxidant capacity. In addition, this biological system also has a large capacity for continuous biomass production, thus demonstrating its high potential for possible nutraceutical applications.

## 1. Introduction

The genus *Ocimum* (Family *Lamiaceae*) includes almost 150 species of herbs and shrubs that inhabit the semi tropical and tropical regions of America (Central and South), Asia and Africa [1,2]. Since prehistoric times, basil has been used as flavoring agent and for fragrance in food and cosmetics [3]. *Ocimum* species, but also cultivars within the same species, varied considerably in their phytochemical compositions [4]. *Ocimum basilicum* L. cv ‘Thai Basil’ is used as a herbal medicinal product [5]. Indeed, Thai basil is a reservoir of a pharmaceutically important metabolite, such as rosmarinic acid and chicoric acid [5,6,7]. These plant specialized metabolites play a pivotal role in plant–environment interactions, provide biotic and abiotic stress tolerance and are used as pharmaceutical agents because they have a variety of antioxidant, anticancer, antimicrobial or anti-diabetic activities [8,9,10,11].

Plant specialized metabolites of pharmacological significance may be difficult to extract in high, sustainable and continuous quantities from field-grown plant material, but in vitro cultures have been identified as a possible biological system for the production of these metabolites [12]. In vitro culture techniques provide consistent production schemes for metabolites that are free from environmental limitations and ensure increased biomass and a high concentration of compounds compared to natural production systems [13]. In addition, in the case of basil varieties or cultivars which can be differentiated on the basis of color, aromatic composition and growth habitat, sometimes harvested from nature, thus complicating its botanical identity, the in vitro system presents a major advantage for the certification of plant authentication [4,14]. Various reports are available in the literature on the in vitro production of bioactive metabolites from the callus culture of medicinally important plants. For example, Smith [15] reported on the production of shikonin by *Lithospermum erythrorhizon* Siebold & Zucc., a natural dye used in food and/or cosmetic applications, as well as of the anticancer drug paclitaxel by *Taxus* L. species or the antimicrobial berberine by *Coptis japonica* Wall cultures. 

Previously, different varieties or cultivars of *O. basilicum* L. have been used to develop different types of in vitro cultures [11,16,17], especially on purple basil [11,18,19]. Moreover, the most promising results obtained with *O. basilicum* in vitro cultures, for high biomass production and high yields of phytochemicals, have been achieved using thidiazuron (TDZ) as a growth regulator. However, TDZ is no longer considered safe in Europe, and therefore its use is excluded for nutraceutical applications, which is the intended application (Status under Reg. (EC) No 1107/2009 (repealing Directive 91/414/EEC)). Here, other growth regulator combinations were analyzed and, most notably, phytochemical production and antioxidant activity were compared to commercial leaf extracts. The current study aimed at establishing a proficient protocol for Thai basil in vitro cultures that are effective in producing continuously high biomass and accumulating high levels of antioxidants under the treatment of various plant growth regulators (PGRs), either alone or in combination. Culture extracts were further analyzed for total phenolic production, HPLC quantification of chicoric acid, rosmarinic acid, and caffeic acid, as well as antioxidant capacity using both in vitro cell-free (DPPH, ABTS and FRAP assays) and cellular assays. These values were compared with those obtained from commercial leaves for extraction. In the light of these results, the nutraceutical potential of the established Thai basil in vitro cultures is discussed. Our findings clearly illustrate the great significance of the optimized biological system, which also has a wide capacity for continuous biomass production, thus demonstrating its high potential for future nutraceutical applications.

## 2. Results and Discussion

### 2.1. Callus Induction and Biomass Production

Callogenesis from 28 days old leaf explants was evaluated for optimum biomass production in response to various PGRs (6-benzylaminopurine (BAP), gibberellic acid (GA_3_), naphthalene acetic acid (NAA)) treatments either alone or in combination (Appendix A). The morphological aspects of the resulting Thai basil callus on different culture media are shown in Figure 1B. Morphological differences in the color and texture of calluses were noted. Here, callogenesis originated at the cut end of explants after 7 to 12 days in response to all tested PGRs. However, the callogenic frequency was varied in response to all PGRs tested. A high callogenic frequency of 95% was observed for BAP:NAA (5 mg/L:1 mg/L) followed by 92% for BAP:NAA (2 mg/L:1 mg/L). Callogenesis was around 20% to 70% for BAP treatments, but the value increased to 95% when NAA was used in combination with BAP (Appendix A). No callus induction was observed in the MS basal medium without hormones, whereas the initiation of callogenesis was observed with NAA alone, but the calluses rapidly turned brown, without noticeable growth, and eventually died.

Maximum FW (190 g/L) and DW (13.05 g/L) biomasses were produced in the presence of BAP:NAA (5 mg/L:1 mg/L) (Figure 1B), while the lowest FW (31.5 g/L) and DW (3.7 g/L) biomasses were noted for GA_3_ (10.0 mg/L) (Figure 1B). Moreover, the maximal production of FW and DW biomasses was detected for all BAP concentrations (0.25–5.0 mg/L) combined with NAA (Figure 1B).

Earlier reports have shown that exogenous auxins in major angiosperm species are essential growth regulators for callus induction [20]. Cytokinin along with auxin is generally required for maximum callus induction by most plant species. In particular, the need for exogenous cytokinin to maintain the interdependent balance between cytokinin and auxin that promotes cell division has been reported as a significant strategy for the production of calluses [21,22]. Likewise, in numerous other medicinal plant species, the combination of NAA with other PGRs resulted in maximum callogenic frequency [23,24]. However, higher concentrations of applied PGRs may limit the callogenesis, possibly due to the repression of some endogenous hormones [25]. The combination of BAP and NAA has been shown to produce optimum biomass for callus cultures of many medicinal plants such as the endangered species *Aquilaria malaccensis* Lam [26]. This may result from the synergistic interaction between BAP and NAA as observed for the production of *Brassica oleraces* L. callus culture biomass [27]. In our previous studies, on *O. basilicum* L. cv purple basil in vitro cultures, optimal production of biomass was achieved using the PGR thidiazuron (TDZ) [11]. However, TDZ is no longer considered safe in Europe, where its use is banned (Status under Reg. (EC) No 1107/2009 (repealing Directive 91/414/EEC)). So, its use for nutraceutical applications, which is the intended application, is no longer suitable.

### 2.2. Production of Phenolic Compounds

#### 2.2.1. Total Phenolic Content (TPC)

In response to BAP combined with NAA (5 mg/L:1 mg/L), calluses yielded a maximum total phenolic content (TPC) of 132.6 mg/g DW. On the contrary, BAP alone, whatever the concentration applied, resulted in a lower TPC ranging from 42.5 to 70.8 mg/g DW. We noted that for each BAP concentration used, the addition of NAA increased TPC. Minimum TPC values ranging from 21.8 to 37.0 mg/g DW have been detected for GA_3_ treatments (0.25–10.0 mg/L). In addition to GA_3_, NAA also stimulated the accumulation of total phenolics. Interestingly, higher TPC was obtained in extracts from in vitro callus cultures grown under optimal PGRs conditions as compared to the extract from commercial leaves of Thai basil (Table 1).

Interestingly, there was a significant correlation between biomass production and total phenolic production (Figure 2, Appendix A). In line with this observation, the biomass-dependent production of total phenolics has already been reported in different plant systems [28].

Szopa and Ekiert [29] noted that PGRs have a direct impact on the biosynthesis of phenolics in medicinal plants. This result is supported by a number of studies which report that the inclusion of a higher BAP concentration and a lower NAA concentration resulted in the maximum TPC [30,31,32,33,34]. This observation could be related to the stimulation of essential enzymes for the biosynthesis of various phenylpropanoids, such as l-phenylalanine/tyrosine ammonia-lyase (PAL/TAL), in response to several PGR concentrations [32]. In contrast, in accordance with our results, GA_3_ has previously been shown to suppress the production of some phenylpropanoids, such as lignans in flax, in a dose-dependent manner [35].

#### 2.2.2. HPLC-Based Evaluation of Simple Phenolics

The HPLC analysis confirmed treatment with different PGRs of leaf-derived callus culture as a tool for improving the synthesis of the main biologically active phenolic acids of Thai basil (Table 1). All BAP and GA_3_ treatments alone and with NAA exhibit a wide range of metabolite production. Maximum accumulation of chicoric acid (36.8 and 35.8 mg/g DW) obtained for the BAP treatments at concentrations of 2 and 5 mg/L, respectively, in combination with NAA at 1 mg/L (Table 1). By contrast, treatments with BAP alone at varying concentrations resulted in the lowest accumulation of chicoric acid ranging from 13.0 mg/g DW to 17.6 mg/g DW (Table 1). Similarly, an optimum production of rosmarinic acid (7.4 mg/g DW) was observed for combined BAP and NAA (5 mg/L:1 mg/L) treatment, while a minimum production was found for all tested BAP concentrations alone (Table 1). Compared to extract from commercial Thai basil leaves, chicoric acid accumulation was 2.6-fold higher in the extract from an in vitro callus produced under optimized conditions, while rosmarinic acid content was similar in both types of extracts (Table 1). In comparison, the accumulation levels in callus extracts of their biosynthetic precursor, caffeic acid, was much lower than that of chicoric acid and rosmarinic acid (Table 1). The concentration of caffeic acid in extract from in vitro callus cultures was 7.5- to 10-fold lower than in the extract from commercial Thai basil leaves (Table 1).

Chicoric, rosmarinic and caffeic acids have previously been reported in different basil species and varieties [36], in particular in Thai basil [6]. It is generally accepted that the accumulation of different phenolic acids in plant cultures may be modulated by the exogenous application of PGRs [29,35,37,38]. In line with this observation, our results show that the phenolic acids chicoric acid and rosmarinic acid can be modulated by PGRs in vitro callus cultures of Thai basil. Similarly, [39] also revealed an increase in the concentration of chicoric acid in their callus cultures of *Echinacea purpurea* grown on NAA supplemented medium. Similarly, increased production of rosmarinic acid was observed in *O. basilicum* in vitro cultures grown on media supplemented with NAA in combination with other PGRs [40], and in *O. americanum* callus culture grown on BAP in combination with IAA [41]. In the present study, chicoric acid and rosmarinic acid were both quantified and their contents were increased in response to PGR treatment. The lower levels of caffeic acid may be associated with its efficient biotransformation into its derivatives rosmarinic acid and chicoric acid [42,43]. It is important to note that chicoric acid production in our Thai basil callus cultures grown under the optimal conditions is higher than that measured for leaves, but also higher than that reported in the possible agricultural production of other plant resources [44]. Moreover, considering its independence of climatic variations or of the potential losses in yield resulting from crop pests’ attacks, as well as, contrary to agricultural production, the possibility offered by the in vitro culture system of multiplying the harvestings, the feasibility of using Thai basil in vitro cultures for the industrial production of chicoric acid is thus possible.

### 2.3. Antioxidant Activities

#### 2.3.1. In Vitro Cell-Free Antioxidant Assays

Three distinct in vitro cell-free antioxidant assays, namely DPPH, ABTS and FRAP were used to assess the antioxidant capacity of the extracts from Thai basil callus cultures. FRAP represents electron transfer (ET) antioxidant mechanism, while ABTS follows hydrogen atom transfer (HAT) antioxidant mechanism. Here, the results of both assays were calculated as TEAC (μM of Trolox C equivalent antioxidant capacity). DPPH assay can reveal both HAT- and ET-based antioxidants, and was calculated as a percentage of DPPH radical scavenging activity (RSA percentage). Extracts from callus grown on BAP:NAA (5 mg/L:1 mg/L) presented the highest levels for DPPH (93.2% RSA) and ABTS (1322.0 µM TEAC) antioxidant assays (Table 2), while lowest levels for DPPH (61.0% RSA) and ABTS (713.1 µM TEAC) were measured for extract from callus grown on 10 mg/L GA_3_ medium (Table 2). Maximum FRAP (523.3 µM TEAC) was measured for extracts from calluses grown on 2 mg/L BAP medium, and lowest value (335.3 µM TEAC) for extracts resulting from calluses grown on 10 mg/L GA_3_ medium (Table 2).

Comparatively, extracts from commercial Thai basil leaves presented a 71.4% RSA for DPPH assay, as well as 914.2 and 387.7 µM TEAC for ABTS and FRAP assays, respectively. Based on these in vitro cell-free assays, the measured antioxidant capacity of the extract from commercial Thai basil leaves was within the range of those determined for callus extracts (Table 2). However, under optimized conditions, extracts from callus cultures showed significantly higher in vitro antioxidant capacity than extract from commercial leaves (Table 2).

Significant correlations between in vitro antioxidant activity and the presence of chicoric acid, rosmarinic acid and caffeic acid were observed for each assay (Figure 2, Appendix A). TPC was also significantly correlated with ABTS antioxidant activity (Figure 2, Appendix A).

Similarly, a significant relationship between phenolic acid accumulation and antioxidant capacity has been revealed for several medicinal plant species [45,46]. From a strictly chemical point of view, these in vitro cell-free antioxidant assays are interesting since they can provide information on the mechanism reaction involved in the antioxidant potential of an extract [47]. Here, the results obtained could reflect a preferential ET-type mechanism. However, these in vitro assays do not necessarily reflect the situation of the in vivo systems and, therefore, their validity should be considered to be strictly limited to interpretation within the meaning of the chemical reactivity with respect to the in vitro produced radicals, and thus should be confirmed by in vivo assays.

#### 2.3.2. Cellular Antioxidant Activity

Antioxidant activity was also evaluated in vivo through the ability of the extract to inhibit the formation of reactive nitrogen and oxygen species (RNS and ROS) using a cellular antioxidant assay (Figure 3). The results confirm the protective action of Thai basil callus extracts against oxidative stress. Maximum inhibition of the ROS/RNS production (56.4%) was measured with extract from callus grown on medium with BAP and NAA (5 mg/L:1 mg/L), while minimum production inhibition ranging from 17.5% to 27.4% was recorded for extracts from calluses grown on media supplemented with BAP alone (Figure 3). 

Extract from commercial Thai basil leaves presented an inhibition of 35.6% of the ROS/RNS production. As a result, extracts from callus cultures grown under optimized conditions also showed significantly higher in vivo antioxidant capacity than extracts from commercial leaves (Table 2), thus demonstrating the interest of this culture system in potential nutraceutical applications. 

The cellular antioxidant assay confirmed the interest of natural antioxidant from plant extracts [18,48,49]. In the past decade, natural antioxidants have generated increasing interest due to their possible use as an alternative to potentially hazardous synthetic antioxidants such as butylated hydroxyanisole (BHA) or butylated hydroxytoluene (BHT) in various food preparations and their possible use as nutraceuticals [50]. Some natural antioxidants have already been shown to be as effective as these synthetic antioxidants [51,52]. In the meantime, plant in vitro systems have been proposed as an efficient alternative for the production of natural antioxidants [53]. The present results indicate the potential use of an in vitro biological system as a tool for continuous and high production of natural antioxidants from Thai basil.

## 3. Materials and Methods

### 3.1. Chemicals and Reagents

In the current research, solvents of analytical grade were used, supplied by Thermo Scientific (Illkirch, France). All reagents and standards were bought from Merck (Saint-Quentin-Fallavier, Lyon, France).

### 3.2. Plant Materials and Callus Culture Establishment

Seeds of Thai basil were obtained from National Agriculture Research Center (NARC, Islamabad, Pakistan). Procedure of Abbasi et al. [23] was adopted for seed surface sterilization. Seeds were immersed for 30 s in mercuric chloride (0.1%) and afterward in ethanol and washed thrice with distilled water. MS basal medium [54] with agar (8 g/L) and sucrose (30 g/L) were utilized for seed germination. For callus culture establishment, 1–1.5 cm long sections of 8 to 10 leaf explant from 28 days old in vitro plantlets were inoculated on MS basal medium with different concentrations of BAP, GA_3_ alone or along with 1 mg/L NAA (Appendix A). Each concentration was replicated three times and the entire experiment was performed twice. Data of callogenic frequency was noted on weekly basis and callus was subcultured with the same PGRs concentration after every 4 weeks. Following this, callus cultures were harvested after the first subculture for the determination of fresh weight (FW) and dry weight (DW) respectively, as described previously by Nazir et al. [11].

In addition, 4 different samples of commercial leaves of Thai basil were purchased from organic markets (NaturéO, Chartres, France) for comparison purpose. 

### 3.3. Sample Extraction

The extracts from fine powder of lyophilized dried sample were obtained using the validated green extraction method designed for the extraction of rosmarinic acid derivatives from *Lamiaceae* [55]. Shortly, the extraction consisted in S/M ratio of 25:1 mL/g DW mixed with 20 mL of pure ethanol (100%) as the extraction solvent extracted during 45 min in an ultrasound bath (USC1200TH, Prolabo, Sion, Switzerland) with operating at an ultrasound frequency of 30 kHz with extraction temperature set at 45 °C. After extraction, centrifugation was carried out for 15 min at 8000× *g*, and the resulting supernatant filtered (0.45 µm, Merck, Saint-Quentin-Fallavier, Lyon, France) prior to further analysis.

### 3.4. Determination of Total Phenolic (TPC)

Determination of TPC was performed by adopting the procedure of Singleton and Rossi [56]. Extract (20 µL) was mixed with Folin-Ciocalteu (FC) reagent (Merck, Saint-Quentin-Fallavier, Lyon, France) (90 µL) and kept for 5 min before adding sodium carbonate (90 µL). Gallic acid as a positive and pure ethanol (extraction solvent) as a negative control were used. Absorbance was recorded at 630 nm with microplate reader. By using the following formula, the total phenolic production was estimated: TPP (mg/L) = TPC (mg/g) × DW (g/L).

### 3.5. HPLC Analysis

Chicoric acid, rosmarinic acid and caffeic acid were estimated by HPLC using a method validated according to international standards of the association of analytical communities (AOAC) [11]. A reversed phase HPLC fitted with autosampler, Varian (Les Ulis, France) Prostar 230 pump and a Varian Prostar 335 photodiode array detector was utilized by following previously reported protocol [57]. A RP-18 Purospher (Merck, Saint-Quentin-Fallavier, Lyon, France) column (250 mm × 4.0 mm, internal diameter: 5 μm) was used at 35 °C for separation. The mixture of acetic acid (0.2%), water and methanol were used as mobile phase. During a 60 min run, the composition of the mobile phase was varied linearly from 5:95 (*v*/*v*) to 100:0 (*v*/*v*) at a uniform flow rate of 0.6 mL/min. Compounds were detected at 320 and 520 nm. Compound identification was achieved by comparing with their commercial standards (Merck, Saint-Quentin-Fallavier, Lyon, France). Quantification was carried out using 5-point calibration curves 1 with at least 0.998 correlation coefficient [11]. Triplicate assays were conducted, and results were demonstrated as mg/g DW of the sample.

### 3.6. Antioxidant Activity 

#### 3.6.1. DPPH Assay

Free radical DPPH (2,2-diphenyl-1-picrylhydrazyl) scavenging potential was evaluated by following the protocol of Shah et al. [58]. Briefly, the sample extract (20 μL) was blended with 180 μL of DPPH solution and kept for 1h in dark. Using a microplate reader (Synergy II reader, BioTek Instruments, Colmar, France), the absorbance was recorded at 517 nm. Free radical scavenging potential was estimated by following equation: % scavenging = 100 × (1 − AE/AD), with AE = dilution absorbance at 517 nm and AD = pure DPPH absorbance as a standard.

#### 3.6.2. ABTS Assay 

The protocol of Velioglu et al. [59] was adapted for the ABTS (2,2-Azinobis (3-ethylbenzthiazoline-6-sulphonic acid) assay. Briefly, by combining equal amounts of ABTS salt (7 mM) and potassium per sulphate (2.45 mM), a solution was prepared, then placed for 16 h in the dark. The sample extract (10 µL) was blended with the ABTS solution (190 µL). Absorbance was measured at 734 nm using a microplate reader (Synergy II reader, BioTek Instruments, Colmar, France), after keeping the mixture for 15 min in the dark at 25 °C. Triplicate assays were performed and antioxidant power was expressed as Trolox equivalent antioxidant capacity (TEAC).

#### 3.6.3. Ferric Reducing Antioxidant Power (FRAP) Assay 

To find out FRAP, the protocol of Benzie and Strain [60] was adapted. The sample extract (10 µL) was blended with the FRAP solution (190 µL). Absorbance was recorded at 630 nm using a microplate reader (Synergy II reader, BioTek Instruments, Colmar, France), after incubating the mixture at 25 °C for 15 min. Triplicate assays were performed and antioxidant power was demonstrated as TEAC.

#### 3.6.4. Cellular Antioxidant Assay

The cellular antioxidant assay was performed with the detection of reactive nitrogen (RNS) and oxygen (ROS) species levels were determined by using a fluorescent dye dihydrorhodamine-123 (DHR-123) as described previously by Garros et al. [61]. All extracts evaporated under nitrogen flow, dissolved in DMSO at 50 µg/mL, and added to the cells 6 h before oxidative stress induction at a final concentration of 1 mg/mL. The final concentration of DMSO applied on the cell was 1% (*v*/*v*). For the control sample, DMSO totaling 0.1% of the final volume was added. Overnight incubation of yeast cells with Thai basil extracts was performed. Then, the extracts were washed two times with PBS and kept at 30 °C in the dark for 10 min. The BioRad (Marnes-la-Coquette, France) Fluorimeter (λem = 535 nm, λex = 505 nm) was used for the detection of the fluorescent signal after repeatedly washing with PBS.

### 3.7. Statistical Analysis 

Means and standard deviations of three to six independent replicates were used to present the data. Statistical analysis was performed with XLSTAT 2019 (Addinsoft, Paris, France) following the manufacturer’s instructions using the Kruskal–Wallis test. Significant threshold at *p* < 0.05 was used for all statistical tests and was represented by different letters in the tables and figures. Correlation analysis was performed with Past 3.0 (Øyvind Hammer, Natural History Museum, University of Oslo, Oslo, Norway) using the Pearson parametric correlation test and visualized using Heatmapper [62].

## 4. Conclusions

*Ocimum basilicum* L. cv ‘Thai Basil’ is a widely used herbal medicinal product, particularly due to its accumulation of a pharmaceutically important metabolite, including rosmarinic acid and chicoric acid. Here, a protocol is proposed for the effective production of leaf-derived callus cultures of Thai basil with sustainable and elevated productions of both biomass and antioxidant phenolics. We proved that plant cell and tissue culture technologies can be applied routinely to Thai basil as an important, sustainable and low-cost biomass source for the production of high-value antioxidant compounds. MS basal medium with BAP:NAA (5 mg/L:1 mg/L) yields a maximum biomass accumulation (both fresh weight (FW): 190 g/L and dry weight (DW): 13.05 g/L) as well as enhanced phenolic (346.08 mg/L) production. HPLC quantification analysis confirmed high productions of both chicoric acid (35.77 mg/g DW) and rosmarinic acid (7.35 mg/g DW) under this optimal callus culture conditions. Antioxidant potential, assessed using both in vitro cell-free and in vivo cellular antioxidant assays, revealed maximum in vitro antioxidant activity with both DPPH (93.2% of radical scavenging activity) and ABTS (1322 µM Trolox equivalent antioxidant capacity) assays for the corresponding extracts. In vivo cellular antioxidant activity assay confirmed this effective protection against oxidative stress with an effective inhibition of ROS and RNS production conferred by the corresponding extract. Under optimal conditions, compared to commercial leaves, the extracts of callus showed higher production of chicoric acid, and similar levels of rosmarinic acid, associated with higher antioxidant capacity. Furthermore, considering the large production capacity of this biological system, suitable for continuous biomass and antioxidant phenolic productions, can be further expanded in the future with the use of cell suspension cultures growing in bioreactors, we anticipate that it could have a great potential for possible nutraceutical applications.

## Figures and Tables

**Figure 1 molecules-25-04859-f001:**
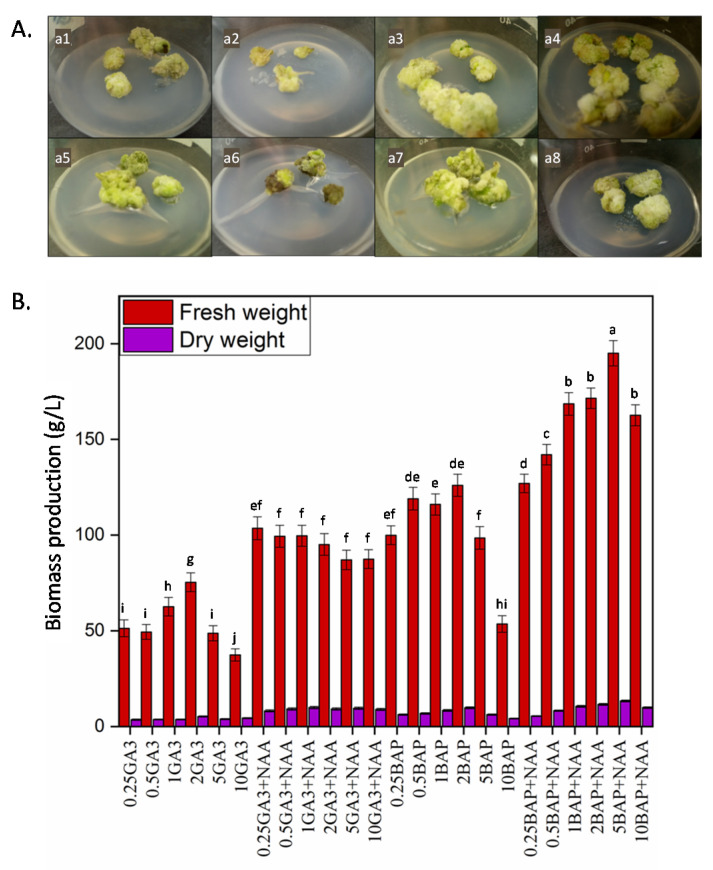
(**A**) Effects of different plant growth regulators (PGRs) on callogenesis from Thai basil leaves explants: (**a1**) 1 mg/L 6-benzylaminopurine (BAP), (**a2**) 5 mg/L BAP, (**a3**) BAP: naphthalene acetic acid (NAA) (0.5 mg/L:1 mg/L), (**a4**) BAP:NAA (5 mg/L:1 mg/L), (**a5**) 2 mg/L gibberellic acid (GA_3_), (**a6**) 5 mg/L GA_3_, (**a7**) GA_3_:NAA (0.25 mg/L:1 mg/L), (**a8**) GA_3_:NAA (5 mg/L:1 mg/L). (**B**) Biomass accumulation in response to different PGRs. Fresh weight (FW) and dry weight (DW) from leaf derived callus grown on BAP (0.25, 0.5, 1, 2, 5, 10 mg/L) and GA_3_ (0.25, 0.5, 1, 2, 5, 10 mg/L) alone or in combination with NAA (1 mg/L). Values are means ± standard deviations (SD) of at least 3 independent experiments. Different letters represent significant differences between the various experimental conditions (*p* < 0.05). Note that no differences were observed for the statistical groups for the FW and DW values.

**Figure 2 molecules-25-04859-f002:**
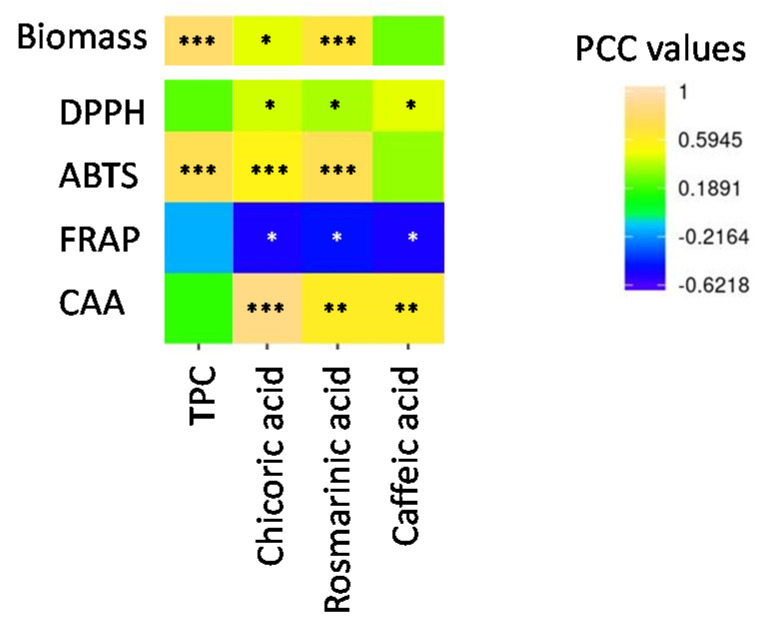
Pearson correlation analysis (PCC) of the relation between the main phytochemicals (TPC: total phenolics content; chicoric acid, rosmarinic acid and caffeic acid) in extracts from Thai basil callus cultures and the different antioxidant assays (in vitro cell-free: DPPH, ABTS, FRAP and in vivo: cellular antioxidant assay (CAA)). *** significant *p* < 0.001; ** significant *p* < 0.01; * significant *p* < 0.05; actual PCC values are indicated in Appendix A.

**Figure 3 molecules-25-04859-f003:**
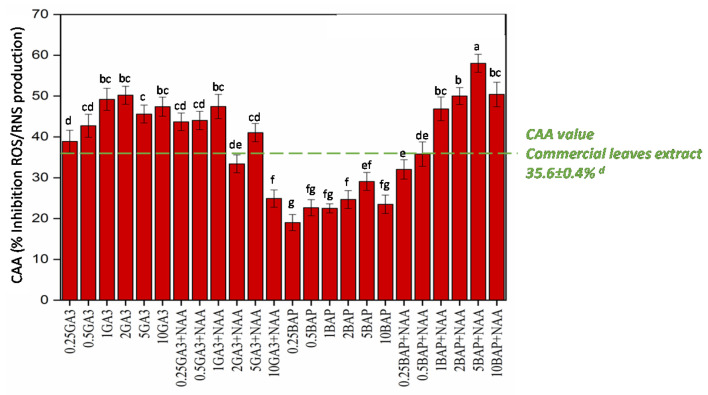
In vivo antioxidant potential of extracts from Thai basil callus cultures in response to different PGRs treatment and leaves measured by cellular antioxidant assay (CAA) estimated as inhibition % of ROS/RNS production. Values are means ± SD of three independent replicates. Different letters represent significant differences between the various experimental conditions (*p* < 0.05).

**Table 1 molecules-25-04859-t001:** Quantification of the main phenolic compounds accumulated in extracts from Thai basil callus cultures in response to different PGRs treatment and leaves.

PGR Treatments	TPC ^1^ (mg/g DW)	Chicoric Acid (mg/gDW)	Rosmarinic Acid (mg/g DW)	Caffeic Acid (mg/g DW)
0.25GA3	27.0 ± 0.9 ^k^	20.8 ± 0.9 ^f^	2.2 ± 0.1 ^f^	0.06 ± 0.03 ^b^
0.5GA3	31.8 ± 1.5 ^k^	23.8 ± 0.7 ^e^	3.1 ± 0.2 ^e^	0.06 ± 0.02 ^b^
1GA3	31.8 ± 3.4 ^k^	28.1 ± 0.7 ^c,d^	3.0 ± 0.2 ^e^	0.06 ± 0.02 ^b^
2GA3	37.0 ± 2.8 ^j^	28.5 ± 0.6 ^c^	3.7 ± 0.2 ^d^	0.07 ± 0.02 ^b^
5GA3	32.7 ± 3.2 ^k^	27.0 ± 0.5 ^d^	3.9 ± 0.2 ^d^	0.06 ± 0.03 ^b^
10GA3	21.8 ± 1.8 ^l^	21.5 ± 0.4 ^f^	3.1 ± 0.2 ^e^	0.05 ± 0.03 ^b^
0.25GA3 + NAA	52.7 ± 3.6 ^i^	22.7 ± 0.4 ^e,f^	4.3 ± 0.3 ^d^	0.06 ± 0.02 ^b^
0.5GA3 + NAA	71.7 ± 4.3 ^f,g^	23.7 ± 0.4 ^e^	4.5 ± 0.3 ^c,d^	0.06 ± 0.02 ^b^
1GA3 + NAA	76.3 ± 0.6 ^f^	26.3 ± 0.5 ^d^	5.1 ± 0.3 ^c^	0.07 ± 0.03 ^b^
2GA3 + NAA	79.9 ± 2.1 ^e^	21.6 ± 0.4 ^f^	5.4 ± 0.3 ^b,c^	0.06 ± 0.02 ^b^
5GA3 + NAA	85.4 ± 4.7 ^d,e^	16.7 ± 0.3 ^g^	5.5 ± 0.3 ^b,c^	0.07 ± 0.03 ^b^
10GA3 + NAA	89.9 ± 1.3 ^d^	15.4 ± 0.3 ^h^	4.9 ± 0.3 ^c^	0.08 ± 0.03 ^b^
0.25BAP	42.5 ± 0.8 ^j^	13.6 ± 0.3 ^i^	2.1 ± 0.2 ^f^	0.04 ± 0.01 ^b^
0.5BAP	61.5 ± 2.7 ^h^	15.3 ± 0.3 ^h^	2.3 ± 0.2 ^f^	0.04 ± 0.02 ^b^
1BAP	66.3 ± 5.3 ^g,h^	17.6 ± 0.3 ^g^	2.8 ± 0.2 ^e,f^	0.04 ± 0.02 ^b^
2BAP	70.8 ± 3.9 ^g^	14.6 ± 0.3 ^h^	2.8 ± 0.2 ^e,f^	0.05 ± 0.02 ^b^
5BAP	45.4 ± 4.6 ^i,j^	13.0 ± 0.2 ^i^	2.5 ± 0.2 ^f^	0.04 ± 0.02 ^b^
10BAP	43.5 ± 3.1 ^j^	14.8 ± 0.2 ^h^	2.4 ± 0.2 ^f^	0.04 ± 0.02 ^b^
0.25BAP + NAA	41.8 ± 4.2 ^j^	20.5 ± 0.3 ^f^	4.2 ± 0.3 ^d^	0.06 ± 0.03 ^b^
0.5BAP + NAA	63.6 ± 3.6 ^g,h^	26.4 ± 0.5 ^d^	4.7 ± 0.3 ^c,d^	0.07 ± 0.03 ^b^
1BAP + NAA	92.6 ± 2.7 ^c,d^	31.3 ± 0.8 ^b^	5.9 ± 0.3 ^b^	0.07 ± 0.02 ^b^
2BAP + NAA	105.4 ± 4.4 ^b^	36.8 ± 1.1 ^a^	7.1 ± 0.4 ^a^	0.07 ± 0.02 ^b^
5BAP + NAA	132.6 ± 3.1 ^a^	35.8 ± 1.2 ^a^	7.4 ± 0.7 ^a^	0.07 ± 0.02 ^b^
10BAP + NAA	96.3 ± 2.3 ^c^	32.5 ± 0.9 ^b^	7.0 ± 0.4 ^a^	0.06 ± 0.02 ^b^
LEAVES	87.1 ± 4.1 ^e^	24.2 ± 1.8 ^e^	7.2 ± 0.7 ^a^	0.6 ± 0.1 ^a^

^1^ Total phenolics content; values are means ± SD of three independent replicates. Different letters represent significant differences between the various experimental conditions (*p* < 0.05).

**Table 2 molecules-25-04859-t002:** In vitro cell-free antioxidant potential of extracts from Thai basil callus cultures in response to different PGRs treatment and leaves.

PGR Treatments	DPPH (% RSA)	ABTS (TEAC µM)	FRAP (TEAC µM)
0.25GA3	85.9 ± 1.7 ^c^	891.7 ± 5.4 ^g^	438.2 ± 5.9 ^d,e^
0.5GA3	74.2 ± 1.9 ^e^	907.5 ± 5.9 ^g^	446.5 ± 5.5 ^d^
1GA3	95.1 ± 1.8 ^a^	826.7 ± 6.0 ^h^	432.6 ± 5.4 ^d^
2GA3	86.9 ± 1.8 ^b^	759.4 ± 4.9 ^j^	420.7 ± 5.5 ^e^
5GA3	91.6 ± 1.7 ^a,b^	814.3 ± 4.9 ^h^	385.5 ± 5.3 ^f^
10GA3	61.0 ± 1.8 ^g^	713.1 ± 5.2 ^k^	335.3 ± 5.8 ^h^
0.25GA3 + NAA	69.8 ± 1.7 ^e,f^	1034.1 ± 7.9 ^d^	403.0 ± 5.1 ^f^
0.5GA3 + NAA	72.0 ± 1.8 ^e^	1100.2 ± 7.6 ^c^	445.7 ± 5.3 ^d^
1GA3 + NAA + NAA	64.4 ± 1.6 ^f^	1170.4 ± 6.7 ^b^	450.4 ± 4.9 ^d^
2GA3 + NAA	88.1 ± 1.8 ^b^	1030.5 ± 6.9 ^d^	499.7 ± 5.2 ^b^
5GA3 + NAA	89.6 ± 1.7 ^b^	933.3 ± 6.9 ^f^	451.4 ± 5.3 ^d^
10GA3 + NAA	79.2 ± 1.5 ^d^	852.8 ± 7.6 ^h^	349.3 ± 4.1 ^h^
0.25BAP	63.6 ± 1.6 ^f^	788.7 ± 7.7 ^i^	483.2 ± 4.9 ^c^
0.5BAP	67.4 ± 1.6 ^f^	784.2 ± 7.5 ^i^	505.5 ± 5.2 ^a,b^
1BAP	73.5 ± 1.5 ^e^	1101.1 ± 7.7 ^c^	496.4 ± 5.4 ^b,c^
2BAP	85.1 ± 2.1 ^c^	878.5 ± 6.8 ^g^	523.3 ± 6.1 ^a^
5BAP	77.8 ± 2.2 ^d^	897.4 ± 7.1 ^g^	517.3 ± 5.7 ^a^
10BAP	72.5 ± 1.9 ^e^	862.0 ± 6.8 ^h^	484.4 ± 4.9 ^c^
0.25BAP + NAA	74.7 ± 2.0 ^e^	849.7 ± 6.4 ^h^	502.9 ± 6.1 ^b^
0.5BAP + NAA	82.6 ± 2.1 ^c,d^	991.2 ± 6.9 ^e^	454.6 ± 4.8 ^d^
1BAP + NAA	79.1 ± 1.9 ^d^	997.3 ± 6.9 ^e^	447.6 ± 4.4 ^d^
2BAP + NAA	82.6 ± 2.1 ^c,d^	1194.3 ± 7.9 ^b^	415.2 ± 4.3 ^e^
5BAP + NAA	93.2 ± 2.2 ^a^	1322.0 ± 7.7 ^a^	384.7 ± 4.1 ^f^
10BAP + NAA	87.2 ± 2.1 ^b^	1080.1 ± 6.9 ^c^	376.7 ± 4.3 ^g^
LEAVES	71.4 ± 3.2 ^e^	914.2 ± 7.3 ^f^	387.7 ± 8.2 ^f^

Values are means ± SD of three independent replicates. Different letters represent significant differences between the various experimental conditions (*p* < 0.05).

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
