# Peer review of "Callus Culture of Thai Basil Is an Effective Biological System for the Production of Antioxidants"

_molecules, 2020, doi:10.3390/molecules25204859_

Round 1

Reviewer 1 Report

Dear all,

below are my comments and suggestions

Manuscript ID: molecules-952997

Title: Callus Culture of Ocimum basilicum L. cv 'Thai Basil' is an Effective Biological System for the Production of Antioxidants Compared to Leaves

In this study, the authors report the possibility of developing a protocol of Thai basil leaf derived callus cultures with sustainable and high production of biomass.

But, in the research database, many protocols are developed on callus cultures.

However, the novelty of the work is to evidence the production continuously high biomass and antioxidants as an alternative of leaves production.

Point 1. Abstract: no changes necessary.

Point 2. Please indicate the international regulations regarding the plant growth regulators (PGRs)

Point 3. Section 4. Conclusions can be improve, add more informations

Point 4. The English language should be revised in all the paper (errors in spelling, grammar, and style)

Point 5. Please check the style for citations and references in all the paper.

The number of bibliographic sources is adequate, but less than 27% of the total bibliographic sources are from the last 5 years.

Author Response

Reviewer 1

Manuscript ID: molecules-952997

Title: Callus Culture of Ocimum basilicum L. cv 'Thai Basil' is an Effective Biological System for the Production of Antioxidants Compared to Leaves

 In this study, the authors report the possibility of developing a protocol of Thai basil leaf derived callus cultures with sustainable and high production of biomass.

But, in the research database, many protocols are developed on callus cultures.

However, the novelty of the work is to evidence the production continuously high biomass and antioxidants as an alternative of leaves production.

AUTHORS: Thank you very much for your very useful comments, we have revised our manuscript accordingly.

 Point 1. Abstract: no changes necessary.

AUTHORS: Thank you very much.

Point 2. Please indicate the international regulations regarding the plant growth regulators (PGRs)

AUTHORS: The regulation ID (Status under Reg. (EC) No 1107/2009 (repealing Directive 91/414/EEC)) has been added in the revised version.

Point 3. Section 4. Conclusions can be improve, add more informations

AUTHORS: The conclusion has been rewritten accordingly.

Point 4. The English language should be revised in all the paper (errors in spelling, grammar, and style)

AUTHORS: A colleague from Leicester University (UK) has checked it.

Point 5. Please check the style for citations and references in all the paper.

AUTHORS: Thank you for this remark. Citations and reference styles have been revised accordingly.

The number of bibliographic sources is adequate, but less than 27% of the total bibliographic sources are from the last 5 years.

AUTHORS: Thank you for this remark. Now, of the 62 cited references, 19 references (30.6 %) were published after 2015, 22 references (35.5 %) after 2014 (considering that 2020 is not finished) and 34 references (54.8 %) were published in the last 10 years. Considering that original papers for some techniques for example have to be cited, we considered that this ratio is now acceptable.

Reviewer 2 Report

The paper presents the results of research on the effectiveness of using callus culture of Thai basil for the production of antioxidant compounds. The work is interesting and well written. 

Due to the possible use of chicoric acid in the pharmaceutical industry, please add information whether the production of these compounds in this system is economically effective.

In the paper, please change the table number on page 7, it should be Table 2 instead of Table 3.

Author Response

Reviewer 2

The paper presents the results of research on the effectiveness of using callus culture of Thai basil for the production of antioxidant compounds. The work is interesting and well written. 

Due to the possible use of chicoric acid in the pharmaceutical industry, please add information whether the production of these compounds in this system is economically effective.

AUTHORS: Thank you very much for your appreciation. We have revised our manuscript according to your remark.

In the paper, please change the table number on page 7, it should be Table 2 instead of Table 3.

AUTHORS: Thank you. It has been revised accordingly.

Reviewer 3 Report

Comments and Suggestions for Authors

Lines 124, 140, 223, and 236 – No “various extraction conditions” are compared in the Figures. The legend should be corrected.

Line 124 – Is it true that no differences were observed for the statistical groups of the FW and DW values? It should be explained or corrected.

Line 260 – The number of leaf explants used for callogenesis is missing, i.e. how many explants were used for calculation of callus induction frequency for each plant growth regulator concentration in the Table S1. The information should be added.

Line 280 – Why was DMSO used as a negative control?

Lines 307 and 308 – The mixtures of callus extracts with reagents were kept for 16 h or for 15 min? What about ratio of extracts and reagents?

Line 318 – The amount of the extract is missing.

Line 323 – “…of three to three separate replicates…” – It should be corrected.

Author Response

Reviewer 3

AUTHORS: Thank you very much for your very useful comments, we have revised our manuscript accordingly.

Lines 124, 140, 223, and 236 – No “various extraction conditions” are compared in the Figures. The legend should be corrected.

AUTHORS: Thank you. It has been revised to “various experimental conditions”.

Line 124 – Is it true that no differences were observed for the statistical groups of the FW and DW values? It should be explained or corrected.

AUTHORS: Yes, no difference was observed for the composition of the statistical groups between FW and DW values. This could be explained by the low variation in moisture content between each experimental condition (various PGR, not known to have an impact on cellular water content, unlike ABA for example) here observed.

Line 260 – The number of leaf explants used for callogenesis is missing, i.e. how many explants were used for calculation of callus induction frequency for each plant growth regulator concentration in the Table S1. The information should be added.

AUTHORS: For each condition, 8 to 10 initial explants were used. This information has been included to the Material and Method section (subsection 3.2).

Line 280 – Why was DMSO used as a negative control?

AUTHORS: This is a mistake during the writing of this subsection. Pure ethanol (extraction solvent) was used as negative control for in vitro antioxidant assays. It has been revised. Thank you very much for your carefulness. For cellular antioxidant assay, DMSO was used as negative control.

Lines 307 and 308 – The mixtures of callus extracts with reagents were kept for 16 h or for 15 min? What about ratio of extracts and reagents?

AUTHORS: Thank you for this comment. This subsection was indeed unclear. It has been rewritten: “All extracts evaporated under nitrogen flow, dissolved in DMSO at 50 µg/mL, and added to the cells 6 h before oxidative stress induction at a final concentration of 1 mg/mL. The final concentration of DMSO applied on the cell was 1 % (v/v). For the control sample, DMSO to 0.1% of the final volume, was added.”

Line 318 – The amount of the extract is missing.

AUTHORS: Thank you for this comment. This subsection was indeed unclear. It has been rewritten, and extract concentration is now provided: “All extracts evaporated under nitrogen flow, dissolved in DMSO at 50 µg/mL, and added to the cells 6 h before oxidative stress induction at a final concentration of 1 mg/mL. The final concentration of DMSO applied on the cell was 1 % (v/v). For the control sample, DMSO to 0.1% of the final volume, was added.”

Line 323 – “…of three to three separate replicates…” – It should be corrected.

AUTHORS: Thank you for this remark. It has been corrected: “Means and standard deviations of three to six independent replicates were used to present the data.”.

Reviewer 4 Report

The article requires a few small corrections.

  • The citations in the text should be equally quoted throughout the article (both in the Introduction and in the rest of the text, e.g. in the discussion of the results; e.g. in italics or not in italics).
  • Please check the text for spaces, especially between numeric value and unit.
  • Black rows in the tables - unnecessary treatment.
  • Line 266: "... as described previously [10]". It would be better to write "... as described previously by Nazir et. al. (2019)". Please check the entire text in this respect.
  • Line 289: Please give the data of the producer of the column (city, country).
  • Please provide the model of the apparatus used in spectrophotometric determinations, its manufacturer and its data (city, country).
  • Please standardize the unit symbol "liter" - "l" or "L" (ml, mL, μl, μL) throughout the article. Currently, Authors use "l" once and "L" other times.

Author Response

Reviewer 4

AUTHORS: Thank you very much for your very useful comments, we have revised our manuscript accordingly.

The article requires a few small corrections.

  • The citations in the text should be equally quoted throughout the article (both in the Introduction and in the rest of the text, e.g. in the discussion of the results; e.g. in italics or not in italics).

AUTHORS: Thank you for this remark. Citations and reference styles have been revised accordingly.

  • Please check the text for spaces, especially between numeric value and unit.

AUTHORS: Thank you for this remark. It has been revised accordingly.

  • Black rows in the tables - unnecessary treatment.

AUTHORS: Thank you for this remark. It has been revised accordingly.

  • Line 266: "... as described previously [10]". It would be better to write "... as described previously by Nazir et. al. (2019)". Please check the entire text in this respect.

AUTHORS: Thank you for this remark. It has been revised accordingly.

  • Line 289: Please give the data of the producer of the column (city, country).

AUTHORS: Thank you for this remark. The information is now provided.

  • Please provide the model of the apparatus used in spectrophotometric determinations, its manufacturer and its data (city, country).

AUTHORS: Thank you for this remark. The information is now provided.

  • Please standardize the unit symbol "liter" - "l" or "L" (ml, mL, μl, μL) throughout the article. Currently, Authors use "l" once and "L" other times.

AUTHORS: Thank you for this remark. It has been revised accordingly.

Round 2

Reviewer 3 Report

no more